# CLEVR-Implicit: A Diagnostic Dataset for Implicit Reasoning in Referring Expression Comprehension

**Jingwei Zhang**[1,*], **Xin Wu**[1,*], **Yi Cai**[1,2,†]

[1]School of Software Engineering, South China University of Technology
[2]Key Laboratory of Big Data and Intelligent Robot (South China University
of Technology) Ministry of Education
ycai@scut.edu.cn

## Abstract

Recently, pre-trained vision-language (VL) models have achieved remarkable success in various cross-modal tasks, including referring expression comprehension (REC). These models are pre-trained on the large-scale image-text pairs to learn the alignment between words in textual descriptions and objects in the corresponding images and then fine-tuned on downstream tasks. However, the performance of VL models is hindered when dealing with implicit text, which describes objects through comparisons between two or more objects rather than explicitly mentioning them. This is because the models struggle to align the implicit text with the objects in the images. To address the challenge, we introduce CLEVR-Implicit, a dataset consisting of synthetic images and corresponding two types of implicit text for the REC task. Additionally, to enhance the performance of VL models on implicit text, we propose a method called Transform Implicit text into Explicit text (TIE), which enables VL models to process with the implicit text. TIE consists of two modules: (1) the prompt design module builds prompts for implicit text by adding masked tokens, and (2) the cloze procedure module fine-tunes the prompts by utilizing masked language modeling (MLM) to predict the explicit words with the implicit prompts. Experimental results on our dataset demonstrate a significant improvement of 37.94% in the performance of VL models on implicit text after employing our TIE method.

## 1 Introduction

In recent years, pre-trained vision-language models have demonstrated remarkable achievements in various vision-language downstream tasks (Lu et al., 2019; Su et al., 2019; Kim et al., 2021; Yao et al., 2021; Kamath et al., 2021; Liu et al., 2022), including referring expression comprehension (REC).

---

[*]Equal contribution.
[†]Corresponding author.

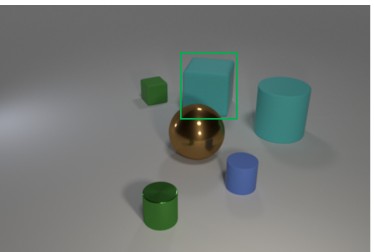

**The left one of the only two objects with the same color and material**

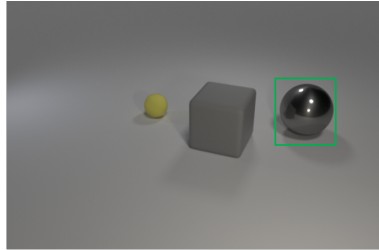

**The right one of the only two objects with different shape and material**

Figure 1: Examples from CLEVR-Implicit. There are two types of implicit text: Same and Different.

REC involves identifying a specific object in an image based on corresponding textual descriptions. A common approach is to pre-train a vision-language model on large-scale image-text pairs to learn the alignment between words in textual descriptions and objects in the images. Fine-tuning is then performed to optimize model performance (Lu et al., 2019; Su et al., 2019; Kamath et al., 2021).

Despite the state-of-the-art results demonstrated by these methods, they still exhibit limitations. Pre-trained models are good at processing explicit text, which describes the appearance attributes of the objects in the image, whether they are reference or target objects. For example, consider the phrase "The object left on the red cube.", models rely on the explicit word "red cube" to locate the referred object in the image. However, they encounter challenges when it comes to implicit reasoning, which is processing the implicit text in REC. Implicit

reasoning occurs when the textual expression indirectly describes an object through a comparison between two or more objects, one type of implicit text is formatted like "The left one of two same color objects". The phrase "two same color objects" does not describe the appearance attributes of any objects, only the relationships between them, and there is no direct reference for the models to infer the target. Consequently, models struggle to align "two same color objects" with any objects in the image. To evaluate the performance gap between explicit and implicit text, we conducted experiments using existing methods on both types of text separately. Preliminary results indicate that the performance on implicit text is approximately 42% lower compared to explicit text.

Besides, the implicit text also plays a significant role in the domain of human-computer interaction. However, existing referring expression datasets, such as RefCOCO (Yu et al., 2016), RefCOCO+ (Yu et al., 2016), RefCOCOg (Mao et al., 2016), and CLEVR-Ref+ (Liu et al., 2019), primarily consist of explicit text. Unfortunately, there is currently no dataset available that is specifically designed to provide research data on implicit text.

To address the lack of research data and methods for implicit text, we propose a synthetic dataset called CLEVR-Implicit, specifically for implicit reasoning in REC. Unlike existing datasets, the implicit text we constructed is a subset of all types of implicit text, since they describe synthetic images, and do not need to consider external knowledge. Specifically, We focus on the more difficult types (only containing relationships between objects without any description of their appearance attributes). We evaluate the current pre-trained REC models on our CLEVR-Implicit dataset with these types of implicit text.

In real-world scenarios, most humans will first extract explicit information from visual context and text information when processing these types of implicit text, and then reason the target object from the extracted explicit information. Inspired by the thinking mode of humans, we simulate the process of humans reasoning about the implicit text and introduce a method called TIE (**T**ransform the **I**mplicit text into **E**xplicit text), which enables pre-trained models to process the implicit text by extracting the explicit information. By employing TIE, REC models facilitate alignment between explicit words and corresponding images on CLEVR-

Table 1: The attributes and example values of the objects in CLEVR. We build the CLEVR-Implicit based on these objects.

| Attribute | Example values |
|-----------|----------------|
| color | red,blue,cyan,purple,gray,green |
| material | metal,rubber |
| shape | cube,cylinder,sphere |
| size | large,small |
| spatial | left,right,behind,front |

Implicit. Experimental results demonstrate a substantial difference in the performance of REC models on our dataset before and after applying TIE.

We summarize our contributions as follows:

- We analyze the limitations of existing referring expression comprehension models on the implicit text and construct the CLEVR-Implicit dataset.

- We propose the TIE method to enable pre-trained REC models to process the implicit text by transforming the implicit text to explicit text.

- We evaluate the existing models on the CLEVR-Implicit and analyze the differences between the models before and after applying TIE.

## 2 Related Works

### 2.1 Referring Expression Comprehension

Referring expression comprehension (REC) aims to localize the region in an image that corresponds to a given sentence description. In recent years, there has been a significant increase in research attention towards this task, driven by the increasing applications of REC in the domain of human-computer interaction. The related task of referring expression generation (REG) has been studied for many years (Krahmer and Van Deemter, 2012; Mitchell et al., 2013; Reiter and Dale, 1992) in the natural language processing (NLP) domain.

Several real-world datasets, such as RefCOCO, RefCOCO+, and RefCOCOg, have been proposed based on the MSCOCO dataset (Vinyals et al., 2016) for this task. CLEVR-Ref+ (Liu et al., 2019) is a synthetic dataset that contains geometric objects. These datasets have been widely adopted as benchmarks for evaluating the performance of referring expression models. MMI (Mao et al., 2016)

directly predicted the bounding box by ranking the region proposals. SLR (Yu et al., 2017) utilized a joint embedding model to predict the referred object by learning the visual and language representations in an embedding space. MattNet (Yu et al., 2018) adapted the attention mechanism to parse expressions into three modules and predict the visual regions by a weighted combination of module scores. VL-BERT (Su et al., 2019) adopted a Transformer model and concatenates the visual and language embeddings as inputs. MDETR (Kamath et al., 2021) constructed an end-to-end system based on the DETR (Carion et al., 2020) detector and a language model to learn the alignment between objects and phrases. VLTVG (Yang et al., 2022) improves comprehension ability by utilizing a visual-linguistic verification module to generate discriminative feature representations.

## 2.2 Prompt Learning

Prompt learning, initially introduced in the field of NLP, is used to reformulate downstream tasks to the form of the pre-training task masked language modeling (MLM) (Devlin et al., 2018). This approach enables models to effectively utilize the knowledge during the pre-training stage (Liu et al., 2023). Concretely, prompt engineering reformulates the downstream task into a "fill-in-the-blank" cloze task with the same objective function as in the MLM task. The effectiveness of prompt learning heavily relies on the design of appropriate prompt templates. Even slight variations in the prompt template can yield significant differences in performance. Therefore, the selection of suitable prompt templates becomes crucial for achieving desirable results.

Prompt learning has also been explored in the multi-modal domain through various studies. CLIP (Radford et al., 2021) pre-trained on large-scale image-text pairs and designs manual prompts like "The photo of [MASK]" to facilitate image classification by predicting the masked token. DPT (Liu et al., 2022) reformulated the visual question answering (VQA) task as the MLM by converting the input (question) to a declaration with the masked token.

## 3 Methodology

This section begins by explaining the construction process of the CLEVR-Implicit dataset. Subsequently, we introduce our proposed TIE method,

Table 2: The statistic data of CLEVR-Implicit.

| Attribute num | Same | Different |
|---|---|---|
| 1 | 3000 | - |
| 2 | 3000 | 1553 |
| 3 | 3000 | 3000 |
| 4 | 2905 | 3000 |

which involves converting the implicit text within the CLEVR-Implicit dataset into explicit text. This explicit text is then utilized in the referring expression comprehension task through the fine-tuning process of pre-trained vision-language models.

### 3.1 Dataset Construction

#### 3.1.1 CLEVR-Implicit Overview

CLEVR-Implicit Dataset uses a subset of the CLEVR (Johnson et al., 2017) scenes (10K images for all, 60% for the train set and 40% for the validation and test set). Table 1 provides an overview of the attributes and values associated with the objects in CLEVR. In order to enhance the difficulty of the dataset, we filter and choose the images that contain 3 or more objects. Given that CLEVR-Ref+ provides scene graphs about the attributes (such as color, material, shape, and size) of objects, as well as the spatial relationships between them within the image, we generate corresponding referring expressions for each image. A total of 19K referring expressions are generated, averaging 1.9 referring expressions for each image. Finally, the bounding box of the referred object is assigned as the label for each referring expression in the CLEVR-Implicit Dataset.

#### 3.1.2 Implicit Text Generation

To generate implicit text that describes objects through a comparison between two or more objects, we utilize the scene graphs provided by CLEVR-Ref+. A program is employed to extract the attributes of all the objects present in the images. This allows for the comparison and analysis of their attributes, enabling the generation of implicit text that captures the relationships between objects without explicitly mentioning any specific object in the image.

For an image, we compare the attributes of each two objects and find each set of objects that satisfies the requirement for generating implicit text. In CLEVR-Implicit, we set up two types of implicit text: "same" and "different". The "same"

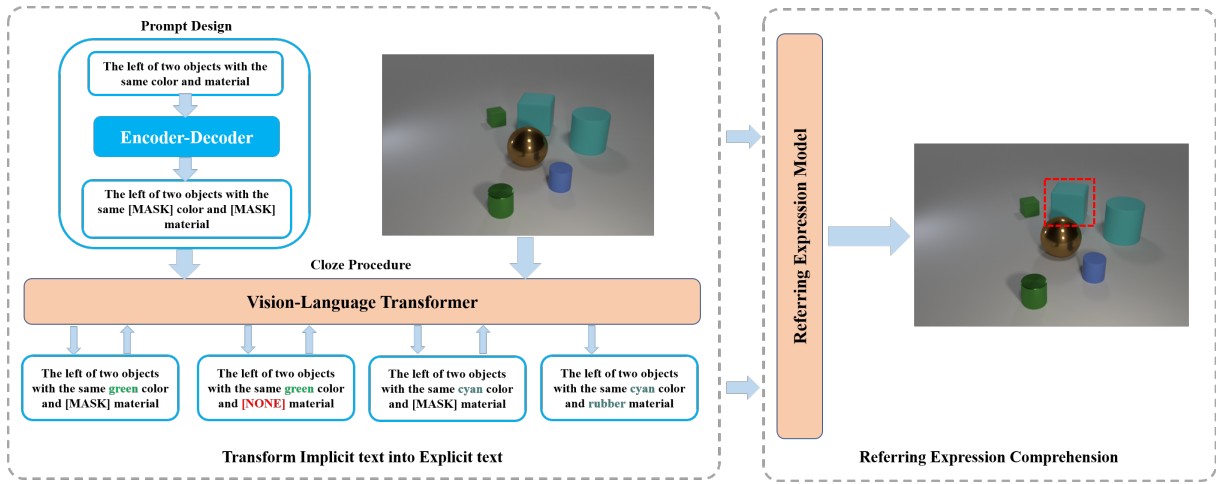

Figure 2: The framework of our TIE method and subsequent REC process. The implicit text in CLEVR-Implicit is converted to an implicit prompt which contains a masked token before the attribute. Then vision-language model predicts the explicit words by Multi-step and Path-select prediction strategies in the MLM task. After TIE, the output explicit text and the image are fed to referring expression model to predict the result.

type represents that the text refers to the object by describing the same of certain attributes between objects (e.g. "The left one of the only two objects with the same color and material"), while the "different" type is the opposite of "same" (e.g., "The left object that is different shape and size from the other one). Table 2 provides statistics regarding the distribution of "same" and "different" type samples in the CLEVR-Implicit dataset.

It is worth mentioning that the "same" type of referring expressions also exist in the CLEVR-Ref+ dataset. The key difference in "same" type text between CLEVR-Implicit and CLEVR-Ref+ is whether the explicit words are hidden. The "same" type of text in CLEVR-Ref+ is like "The objects that have the same size as the red sphere", which contains the explicit word "red sphere". The explicit context provides REC models with a reference object and helps them locate the target. Oppositely, the "same" type of implicit text in CLEVR-Implicit is formatted like "The two objects with the same size", which hides explicit words. Without explicit context and hard to find the reference object, REC models are forced to compare every two objects to identify what the "size" is and whether they are the same. That illustrates that implicit text in CLEVR-Implicit is more challenging.

We pay more attention to the reference to a single object in the context of REC. During the generation process, we take precautions to avoid text ambiguity, ensuring that the implicit text describes only one object in the image. To achieve this, we

prioritize maximizing the number of attributes involved in attribute comparisons. For example, if the target is in a group of objects that share the same color only, we must ensure no other group of objects that can satisfy the same color and material. In cases where such a group of objects exists, we will change the target and modify the implicit text to describe the corresponding number of attributes until no additional attributes can be included. More details of the implicit text generation are shown in Appendix A.

The implicit text in CLEVR-Implicit follows a standardized format based on several templates and rules. Especially, it is common for the same sentence to describe different target objects across different images. By employing homogeneous text, we purposefully direct the model's focus toward the image itself, thereby effectively testing the model's reasoning ability.

## 3.2 TIE Method

### 3.2.1 TIE Overview

To address the challenge of implicit text comprehension, we propose a method called Transform Implicit text to Explicit text (TIE). The objective of TIE is to incorporate explicit words into the original implicit text, enabling the model to reason more effectively. Our approach consists of two key modules: (1) The **prompt design** module builds prompts for implicit text by adding masked tokens. By strategically designing the prompts, we prepare the groundwork for the next module. (2)

The **cloze procedure** module uses a pre-trained vision-language model to predict the explicit words with the implicit prompts.

By leveraging these two modules, TIE enables the integration of explicit words into the implicit text and facilitates indirect reasoning on the implicit text effectively.

### 3.2.2 Prompt Design Module

The prompt design module converts the implicit text into a format suitable for the Masked Language Modeling (MLM) task. One approach to adding explicit words is by obtaining the specific attribute values involved in the comparison of attributes in the implicit text (e.g., "What is the color in 'The left one of the only two objects with the same color'?"). Consequently, the prompt design module generates the prompt like "The left one of the only two objects with the same [MASK] color" for an implicit text sample. It's important to note that the conversion process differs between the "same" and "different" types of implicit text. In the "same" type, only one identical attribute value needs to be predicted, while in the "different" type, two different attribute values must be predicted. Therefore, in the "different" type, two masked tokens are added before each attribute in the implicit text. This process of converting the implicit text can be seen as a translation task, where the source text is the original implicit text, and the target text is the input format required for the MLM task. To accomplish this conversion process, we create an implicit prompt dataset using the implicit text from CLEVR-Implicit as the source text, and the target text is a text with the masked token preceding each attribute in the implicit text. Subsequently, we train an encoder-decoder model (T5, Raffel et al., 2020) on the implicit prompt dataset to add the appropriate number of masked tokens corresponding to the attributes for each sample in CLEVR-Implicit.

### 3.2.3 Cloze Procedure Module

The training objective of the cloze procedure module aligns with the MLM task, a commonly employed pre-training task in various pre-trained models, so another cloze dataset is devised for the MLM training stage. In this dataset, the input consists of implicit text containing masked tokens and an image description, while the target comprises explicit text containing the ground-truth attribute values. In this module, we fine-tune a pre-trained vision-language model ViLT to predict a specific word

within the masked tokens. And we will introduce three prediction strategies: Base, Multi-step, and Path-select.

**Base** In the prediction stage, the prediction of multiple attribute values in a sample is different from the prediction of a single attribute value. For implicit text containing multiple attributes, a straightforward prediction strategy is to predict all attribute values. However, this strategy tends to decrease the correlation among the predicted values, resulting in potential discrepancies where multiple predicted values may not refer to the same group of objects. Consequently, incorrect prediction results may arise.

**Multi-step** To address the aforementioned issue, we propose an enhanced strategy named multi-step prediction. Specifically, when predicting a sample with multiple attributes, the model outputs only one attribute value at a time. Subsequently, it incorporates the previous prediction as input to predict the next attribute value. Compared with the method of predicting all attribute values at one time, this strategy enhances the correlation among multiple attribute values. For instance, if the original text is "The left one of the only two objects with the same [MASK] color and [MASK] material," the corresponding first-step output is adjusted to "The left one of the only two objects with the same red color and [MASK] material." And the second-step output is "The left one of the only two objects with the same red color and metal material."

**Path-select** We find that the Multi-step strategy has a potential challenge where the accuracy of predictions relies on the first attribute. If the first attribute prediction is incorrect, it might trigger a chain reaction leading to subsequent attributes being errors. To mitigate this issue, we search for some objects that are not the targets as negative samples. They are characterized by partially satisfying the constraints of implicit text. Part of the attributes that do not satisfy are marked [NONE]. For instance, for a target sample "The same red color and rubber material" in cloze dataset, the corresponding negative sample might be "The same blue color and [NONE] material".

During the prediction stage, when the model obtains the optimal $Softmax$ result for an attribute and predicts [NONE] for the next attribute, it recognizes this as an incorrect predicted path. Subsequently, the model reverts to the prediction of the preceding attribute, considering the suboptimal

| ViLT (Kim et al., 2021) | Same | | Different | | Total | |
|---|---|---|---|---|---|---|
| MLM | val | test | val | test | val | test |
| Base | 80.54 | 81.97 | 25.73 | 23.35 | 61.99 | 60.48 |
| Multi-step | **92.55** | **91.42** | **87.05** | 85.47 | 91.45 | **91.50** |
| Path-select | 91.76 | 91.35 | 85.58 | **87.50** | **91.48** | 90.37 |

Table 3: The TIE experiment results on CLEVR-Implicit. We use three different prediction strategies to compare the performance of ViLT.

| Method | Before TIE | | After TIE | | Ground-truth | |
|---|---|---|---|---|---|---|
| | val | test | val | test | val | test |
| MDETR (Kamath et al., 2021) | 53.57 | 53.89 | 91.51 | 92.19 | 96.01 | 95.66 |
| VLTVG (Yang et al., 2022) | 58.83 | 58.46 | 84.48 | 86.15 | 95.97 | 96.85 |

Table 4: Referring Expression Comprehension results performed by MDETR and VLTVG. Before TIE means that the models are trained on the implicit text before the input transformation of the TIE method, and After TIE means that the models are trained on the explicit text after TIE.

$Softmax$ result, which guides the model toward the correct prediction.

## 4 Experiments

### 4.1 Implemention Details

For the REC task on the CLEVR-Implicit dataset, we employ two models: MDETR (Kamath et al., 2021) and VLTVG (Yang et al., 2022). MDETR utilizes weights pre-trained on various datasets such as COCO (RefCOCO, RefCOCO+, RefCOCOg), Visual Genome (Krishna et al., 2017), and Flickr30k (Plummer et al., 2015). and VLTVG uses weights pre-trained on RefCOCOg. And we fine-tune them on the CLEVR-Implicit dataset. During the training stage, we use the settings of learning rate 1e-4, learning rate of backbone 1e-4, batch size 16 and weight decay 1e-5 to train for 60 epochs. In the input transformation experiment, we utilize the T5-small model for the prompt design module. As for the cloze procedure module, we select ViLT (Kim et al., 2021). To ensure a fair comparison, all different prediction methods are configured with the AdamW optimizer, a learning rate of 1e-4, a batch size of 32, and an image size of 384x384 to train for 70 epochs.

### 4.2 Results of TIE

Table 3 presents the results of the TIE experiment. The evaluation method is to assess whether the predicted explicit words match the corresponding ground-truth values. For samples containing multiple attributes, it is necessary to ensure the accurate prediction of all explicit words. otherwise, it is considered incorrect. In the case of "different" type

samples, where attribute values differ between objects, the order of the two predicted values for a single attribute can be reversed.

We present the results separately for training on the "same" and "different" types of samples, and the "Total" column represents the combined performance on both types of implicit text. In the Base strategy, we observe that "different" samples are more challenging to understand compared to "same" samples, resulting in a decrease in performance of up to 54.81%. For "different" samples, the model needs to output two values to predict a single attribute, resulting in twice as many explicit words to be predicted compared to "same" samples. In contrast, the Multi-step strategy exhibits significant improvements for both types of text. It shows a 12.01% increase in performance for "same" samples and a remarkable 61.32% increase for "different" samples. A reasonable explanation is that the prediction of each explicit word takes into account all previously predicted results. Multi-step strategy mitigates cases where the initial predictions are correct, but subsequent predictions suffer from errors, leading to more accurate overall predictions.

From the comparing results of the Path-select and Multi-step strategies, we observe some differences when incorporating negative samples into the training set. The Path-select results on the validation set exhibit a slight decrease of 0.79% and 1.47% compared to Multi-step. We speculate that this is because the majority of incorrect positions in wrongly predicted samples tend to occur in the second attribute or later, and the prediction of the first attribute in these samples is generally accurate.

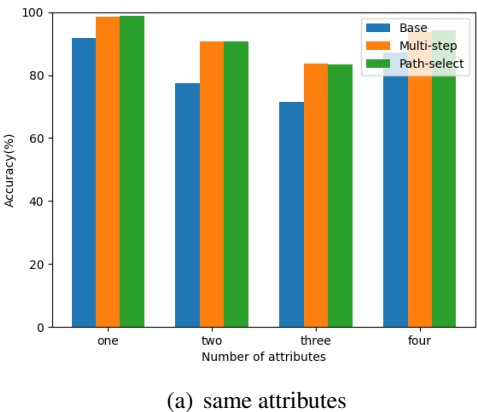 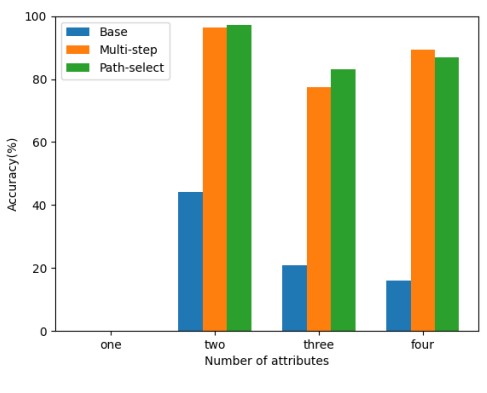

| (a) same attributes | (b) different attributes |

Figure 3: 3(a) is the performance of three prediction strategies on "same" type samples in test set, and 3(b) is the performance of three prediction strategies on "different" type samples in test set.

However, on the test set of "different" samples, Path-select surpasses Multi-step by 2.03% in terms of performance. Particularly, the performance of Multi-step and Path-select in the "Total" column is comparable, indicating that the addition of negative samples can enhance the model's robustness. And negative samples enable the model to choose the correct path when the first attribute prediction is incorrect.

### 4.3 Results of Referring Expression

Table 4 presents the results of the referring expression experiment. We evaluate the performance of MDETR and VLTVG, both before and after applying the TIE based on the Multi-step prediction strategy. Additionally, we provide the performance of the models trained on ground-truth explicit samples as a reference. The evaluation metric is Intersection over Union (IoU), with a threshold value of 0.5. We observe that prior to TIE, the performance of both models trained on implicit text is relatively poor, indicating limited reasoning ability when dealing with implicit text. However, after applying TIE, there is a significant improvement in performance for both models (53.57% vs. 91.51% for MDETR and 58.83% vs. 84.48% for VLTVG). However, compared with Ground-truth, there is a certain gap in performance, because it is limited by the impact of TIE performance.

### 4.4 Results of Contrast Experiment

**Difficulty of CLEVR-Implicit** In order to contextualize the difficulty of CLEVR-Implicit, we evaluate the performance of REC models on CLEVR-Implicit when the models are trained on CLEVR-

| Method | Train on CLEVR-Ref+ | |
| --- | --- | --- |
| | val | test |
| MDETR | 47.02 | 45.85 |
| VLTVG | 47.99 | 47.11 |

Table 5: The performances of REC models which are trained on CLEVR-Ref+ and tested on CLEVR-Implicit.

| VLTVG on CLEVR-Ref+ | val | test |
| --- | --- | --- |
| Before TIE | 66.43 | 65.22 |
| After TIE | 68.05 | 66.46 |

Table 6: The performances of VLTVG respectively on original CLEVR-Ref+ (Before TIE) and CLEVR-Ref+ after inference by TIE trained on CLEVR-Implicit.

Ref+. Table 5 shows that the results of MDETR and VLTVG models are all lower than "Before TIE" Column results in Table4. That indicates that the experience that the models learn from CLEVR-Ref+ cannot help them process the implicit text, because the samples in CLEVR-Ref+ contain explicit information to help the model locate the target, while the samples in CLEVR-Implicit hide the information. Therefore, we believe that it's challenging to process the implicit text in our CLEVR-Implicit dataset.

**Affect of TIE on explicit text** Since the TIE method converts implicit text into explicit text, in addition to implicit text, we also need to analyze whether the explicit text after applying TIE will affect the performance of REC models. To verify this problem, we evaluate the performance of VLTVG respectively on original CLEVR-Ref+ and

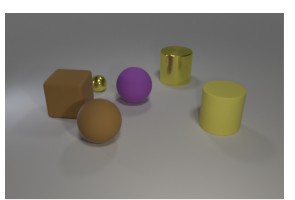

**O**: The left one of the only two objects with the same shape, color and size in the image

**P**: The left one of the only two objects with the same [MASK] shape, [MASK] color and [MASK] size in the image

**GT**: cylinder, yellow, large

**Base**: sphere, brown, large

**Multi-step**: cylinder, yellow, large

**Path-select**: cylinder, yellow, large

**(a) An example of the comparison from Base, Multi-step and Path-select prediction strategies**

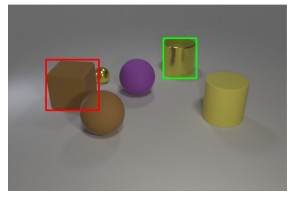

**Before:**
The left one of the only two objects with the same shape, color and size in the image

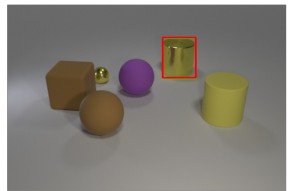

**After:**
The left one of the only two objects with the same cylinder shape, yellow color and large size in the image

**(b) REC before TIE**           **(c) REC after TIE**

Figure 4: Examples from TIE and referring expression comprehension experiments. **(a)** an example of the comparison from Base, Multi-step and Path-select prediction strategies. **O**, **P** and **GT** denote original implicit text, generated prompt text and ground-truth explicit words. **(b)** Visualization of the predicted bounding box from VLTVG before TIE and **(c)** after TIE. green and red rectangles denote ground-truth and predicted bounding box.

CLEVR-Ref+ after inference by TIE trained on CLEVR-Implicit. Table 6 shows that TIE does not affect performance (even better on VLTVG) on other types of referring expressions besides the implicit text, and TIE is robust in a broader class of referring expressions.

**Contrast between explicit and implicit** As shown in Table 7, we evaluate the performance of MDETR and VLTVG models separately on predicted explicit text and implicit text when the models are trained on implicit text and ground-truth explicit text respectively.

When both validation sets provide predicted explicit words (Column 1 vs Column 3), the performance of the model trained on the implicit training set is significantly decreased (MDETR: 31.48% vs 91.98%, VLTVG: 53.85% vs 85.90%) compared with that on the explicit training set, which is evidence that explicit words play a crucial role in the models' comprehension and reasoning processes. The same situation occurs when exchanging implicit and explicit states on train and validation sets (Column 2 vs Column 3).

### 4.5 Performance on different number of attributes

Figure 3 illustrates the performance of ViLT on "same" and "different" samples with different numbers of attributes. In Figure 3(a), we observe that the model achieves the highest performance when predicting one-attribute samples, while the lowest performance is observed when predicting three-

| Method | Train-i Val-pe | Train-ge Val-i | Train-ge Val-pe |
|--------|--------|--------|--------|
| MDETR | 31.48 | 46.25 | 91.98 |
| VLTVG | 53.85 | 48.34 | 85.90 |

Table 7: The contrast experiment for REC. The **i** suffix denotes the set with implicit samples. The **pe** suffix denotes that the explicit words in samples are predicted by TIE. The **ge** denotes the explicit words in samples is ground-truth.

attribute samples. Interestingly, the performance on four-attribute samples is second only to that of one-attribute samples. A reasonable conjecture is that a set of objects is identical when all four properties are the same in CLEVR images, so the model only needs to find two objects that are identical. Figure 3(b) shows that predicting four-attribute "different" samples has the most significant challenge on the Base strategy. However, on the Multi-step strategy, the performance on four-attribute "different" samples surpasses that of three-attribute samples. This indicates that the Multi-step prediction strategy effectively mitigates the impact of an increased number of predicted words, resulting in improved performance.

### 4.6 Case Study

Figure 4 indicates an example of the comparison of different prediction strategies in the TIE experiment, and we visualize the ground-truth bounding box and the bounding box predicted from VLTVG

Table 8: Examples about the application of implicit text in daily life.

| Real-world Implicit Text Examples |
| --- |
| Two persons are doing the same movement. |
| His height is different from each other. |
| His pants and shoes are the same color. |
| The same brand of cosmetics on the table. |
| Take the drugs used for the same symptom. |

before and after TIE in the REC experiment. In Figure 4(a), with the attributes of the example "shape" and "color", the Base strategy mistakenly predicts "sphere" and "brown". We observe that from the image there is a group of objects that is both "sphere" and another group of objects that is both "brown", but there are no such two objects that satisfy "sphere" and "brown" simultaneously. This indicates that the Base strategy performs poorly on multi-attribute samples due to the weak correlation between attributes, leading to the error that different attributes refer to different groups of objects. In contrast, the Multi-step and Path-select strategies enhance the correlation between attributes and make predictions step-by-step, leading to more accurate predictions.

In Figure 4(b), by taking the implicit text as input, VLTVG incorrectly predicts the referred object as a brown cube. While VLTVG predicts the referred object accurately when taking the explicit text after TIE as input. The comparison result indicates that current models are being confused when performing implicit reasoning based on the logical relation conveyed by the implicit text. Only when explicit words are explicitly included, the models can align the explicit words with the corresponding objects in the image for explicit reasoning.

## 5 Implicit Text in Real-world Scenarios

The implicit descriptions in real-world scenarios are more complex and difficult because they not only describe the targets implicitly through the geometrical attributes of color, material, etc., but also contain more attributes as well as external factors. Table 8 shows the implicit text examples in real-world daily life. It is important that even though our generated implicit text relies on the CLEVR images and the provided annotations, our proposed TIE method can still extend the scope beyond the CLEVR domain to real-world scenarios. For example, based on Figure 5 which is chosen from Visual

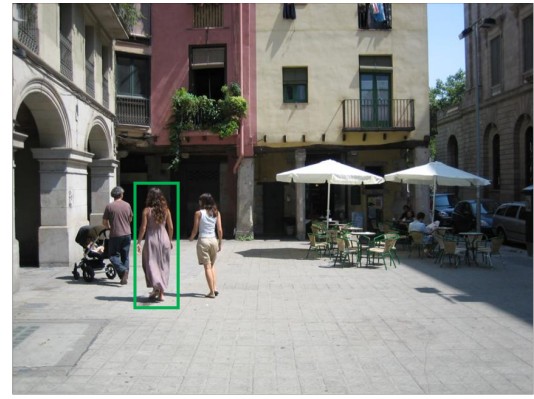

**Original Implicit Text:** The left one of two persons with the same posture and gender

**After Prompt Design Module:** The left one of two persons with the same [MASK] posture and [MASK] gender

**After Cloze Procedure Module:** The left one of two persons with the same walking posture and woman gender

Figure 5: A real-world image from Visual Genome

Genome, consider the implicit text like "The left one of two persons with the same posture and gender". Through the TIE method, the prompt can be built before the "posture" and "gender" by prompt design module, like "The left one of two persons with the same [MASK] posture and [MASK] gender". The prompt guides the cloze procedure module to predict the explicit words "walking" and "woman" like "The left one of two persons with the same walking posture and woman gender". By integrating TIE approach, current models can also process the implicit text on the real-world scenarios.

## 6 Conclusion

In this paper, we analyze the significance of implicit reasoning, along with the lack of investigation in this domain. To address this, we introduce the CLEVR-Implicit dataset for implicit reasoning in the REC task. This dataset utilizes synthesized images to generate two types of implicit text: "same" and "different." Evaluating the performance of existing pre-trained models on CLEVR-Implicit, we observe model confusion when attempting to align implicit text with images. To overcome this challenge, we propose the Transform Implicit text to Explicit text (TIE) method, which converts implicit text into explicit text. Our experiments demonstrate that the reasoning capability of the models improves significantly through the alignment of explicit words and images.

## Limitations

The limitation of our work is that CLEVR-Implicit is built on synthetic images, so it's straightforward to generate corresponding implicit text based on the annotation of the synthetic images. Besides, since the synthetic images contain only geometric objects, our implicit text can only be generated within the scope of the object's attributes. Therefore, in future work, we will annotate more types of implicit text based on real-world scenarios. This will allow us to explore the mechanisms of model understanding of implicit text.

## Acknowledgements

This work was supported by the National Natural Science Foundation of China (62076100), Fundamental Research Funds for the Central Universities, SCUT (x2rjD2230080), the Science and Technology Planning Project of Guangdong Province (2020B0101100002), CAAI-Huawei MindSpore Open Fund, CCF-Zhipu AI Large Model Fund.

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

## A  More Details of Implicit Text Generation

To clarify on our implicit referring expression generation process, we'd like to elaborate on the method used to create the CLEVR-Implicit dataset from the existing CLEVR images and their associated annotations. These annotations contain object attributes, as well as spatial relationships between objects. For each image, we will traverse all the objects in the image by comparing whether the attribute values between two objects are the same/different each time. For example, if an image contains 4 objects, this leads to C(4,2)=6 object pair comparisons (e.g., (1,2), (1,3), (1,4), (2,3), (2,4), (3,4), the numbers are object ids). We will generate the corresponding implicit text if a certain pair satisfies the "same" or "different" condition. Suppose the (2,3) combination objects' shape and color are both cube and blue, and guarantee that no other pairs satisfy the condition "same shape and color" to avoid the ambiguity in the REC task, then we will generate the "same" type of implicit text as "The two objects with the same shape and color". Due to the only one target of the REC task, we must choose one of the (2,3) objects to be the target. We employ a random selection for the target based on the spatial relationship between 2 and 3. If object 2 is left of object 3 and object 2 is selected, the finally generated implicit text is "The left one of the two objects with the same shape and color". The "different" types of text are generated in the same way. This consistent pattern of implicit text generation in the CLEVR-Implicit dataset ensures both the concealment of explicit information and the reference of target objects in the images.