# OpenReview forum: "CLEVR-Implicit: A Diagnostic Dataset for Implicit Reasoning in Referring Expression Comprehension"
_EMNLP/2023/Conference — EMNLP 2023 Main_

### Official Review · Reviewer_rgpy · 2023-08-05

**Soundness:** 4

**Excitement:**

4: Strong: This paper deepens the understanding of some phenomenon or lowers the barriers to an existing research direction.

**Paper Topic And Main Contributions:**

This paper points out the issue that existing vision-language model struggles to deal with implicit text. It constructs a dataset of implicit text matching for related research. To improve the ability of vision-language models to match implicit text, this paper proposes a novel method to utilize masked language models to transform implicit text into explicit text. The proposed method can significantly improve ability of vision-language models on matching implicit text.

**Reasons To Accept:**

1. This paper points out an important issue, implicit text matching, for vision-and-langugae learning. It builds a useful dataset on this task and reveals the weaknesses of existing vision-language models on implicit text matching.
2. This paper proposes a novel and effective methods to transform implicit text into explicit text. In experiments, the proposed method can significantly improve the vision-language models on matching implicit texts.

**Reasons To Reject:**

1. It would be better if the authors can design experiments to discuss the reason why existing vision-language cannot match implicit texts. Does it is because that current vision-language model lacks spatial inference ability or other important ability? (solved by authors in the rebuttal)
2. The authors can also try to fine-tune vision-language models with the TIE, which will better highlight the role of TIE.

**Reproducibility:**

4: Could mostly reproduce the results, but there may be some variation because of sample variance or minor variations in their interpretation of the protocol or method.

**Reviewer Confidence:**

3: Pretty sure, but there's a chance I missed something. Although I have a good feel for this area in general, I did not carefully check the paper's details, e.g., the math, experimental design, or novelty.

---

> ### Author Rebuttal · Authors · 2023-08-29
>
> Thank you for the constructive suggestions on this paper. The comments are helpful to improve our work. We have carefully studied each comment one by one. We will respond point by point to your review and questions about the paper as follow:
>
> **Response-to-Reject1**: Thanks for the constructive comment. Based on our experiment results, the REC performance gap of before and after TIE is very huge, but the only difference is whether explicit information is included in the text. We have also made an additional ablation experiment with trained on CLEVR-Ref+ and tested on CLEVR-Implicit, and get the results in REC task:
>
> |  | val | test |
> | :----: | :----: | :----: |
> | MDETR | 47.02 | 45.85 |
> | VLTVG | 47.99 | 47.11 |
>
> Therefore, we believe that existing models are pre-trained on the text with explicit information so that they can only process the explicit reasoning. But implicit text hides the explicit information and force the models to extract the explicit information by traversing all the objects in images to do the reasoning. However, existing models don’t have this capability. Therefore, we divided the implicit reasoning into two steps: TIE and then REC. By splitting into two steps, our TIE method can be plug-and-play and can be integrated to other referring expression models to do the implicit reasoning.
>
> **Response-to-Reject2**: Thank you for the constructive comment. We will make an attempt to fine-tune VL models with TIE in the future work. We are also exploring an effective end-to-end method to do the implicit reasoning more efficient.

---

### Official Review · Reviewer_WAk9 · 2023-08-07

**Soundness:** 4

**Excitement:**

4: Strong: This paper deepens the understanding of some phenomenon or lowers the barriers to an existing research direction.

**Paper Topic And Main Contributions:**

This paper introduced the CLEVR-Implicit dataset for implicit reasoning in the referring expression comprehension task. In their analysis, they showed that models struggle to align the implicit text with the objects in the images. To address the limitation, they proposed Transform Implicit text into Explicit text (TIE).

**Reasons To Accept:**

1. The idea of the paper is simple but interesting. It shows a limitation of current VLMs on implicit reasoning.
2. The paper reveals that if the implicit text is converted to explicit text, then VLMs perform much better.
3. The paper is well organlzed and easy to follow.


**Reasons To Reject:**

It is a great attempt to evaluate VLM's implicit reasoning on text. However the proposed data is synthetic data and only covered few attributes. It would be great to explore a more realistic setup including diverse attributes.

**Reproducibility:**

3: Could reproduce the results with some difficulty. The settings of parameters are underspecified or subjectively determined; the training/evaluation data are not widely available.

**Reviewer Confidence:**

3: Pretty sure, but there's a chance I missed something. Although I have a good feel for this area in general, I did not carefully check the paper's details, e.g., the math, experimental design, or novelty.

---

> ### Author Rebuttal · Authors · 2023-08-29
>
> Thank you for the constructive comment. We will explore the implicit text in the real-world scenarios currently, which contain diverse attributes. And we will discuss about how the real-world scenarios fit within our TIE framework. Our proposed TIE method can also expand the scope beyond the CLEVR domain to the real-world photographs. For example, based on an Visual Genome [1] image (You can check the image in this anonymous link: https://anonymous.4open.science/r/example_image-C3BD/25.jpg), consider the implicit text like “The left one of two persons with the same posture and gender”. By our TIE method, prompts can be built before the “posture” and “gender”, like “The left one of two persons with the same      posture and      gender”. The prompt guides the model to predict the explicit words “walking” and “woman” like “The left one of two persons with the same walking posture and woman gender”. By integrating TIE approach, current models can also make the implicit reasoning on the real-world scenarios.

---

### Official Review · Reviewer_Gmcn · 2023-08-11

**Soundness:** 4

**Excitement:**

3: Ambivalent: It has merits (e.g., it reports state-of-the-art results, the idea is nice), but there are key weaknesses (e.g., it describes incremental work), and it can significantly benefit from another round of revision. However, I won't object to accepting it if my co-reviewers champion it.

**Paper Topic And Main Contributions:**

The paper studies the problem of grounding referring expressions involving relations between objects (specifically the "same" and "different" relation between different attributes) where the attributes of the object are referred to implicitly. The paper constructs a synthetic dataset in the widely adopted CLEVR framework, with referring expressions generated using templates. Using the dataset, the paper builds referring expression comprehension (REC) models that identify the object in the image being referred to. The paper finds that existing approaches to REC perform poorly on this task, and propose an algorithm that identifies and makes attributes of the objects in the image explicit, which allows existing approaches to reason more effectively. They find that this idea of making information explicit helps overcome the shortcomings of models in this type of reasoning.

**Questions For The Authors:**

- The CLEVR-Ref+ paper includes analysis on expressions that involve the "same" operation. However, there is no discussion of it in this paper. How are their findings different from the findings in this paper? Discussion of this would help better identify the contributions of this paper.
- Why does the prompt design module have to be trainable? At least for the CLEVR domain, it seems like you can use a simple rule to identify words that express an attribute and insert an appropriate number of masked tokens before the word based on whether the expression contains "same" or "different". Is there additional strategy to it?
- In the terms used in Table 6, are the results in Table 5 based on the Train pe/Val pe setting? Making this explicit would help understand when and how TIE is used in training and during inference.
- The discussion of prompt learning in 2.2 is confusing. The wording suggests that the prompts are being learned (improved based on data), but the discussed methods are instances of using prompts to reformulate inputs to make them closer to the model's training distribution, with the changes being performed manually (at least in most of the methods discussed). This section could be improved to be more relevant to the proposed method.
- Additional examples of the "different" samples would also help better understand the construction of that portion of the dataset.

**Reasons To Accept:**

- The paper undertakes a systematic and thorough analysis of how models perform on a specific aspect of reasoning about language about images.
- The paper finds that using models (which may be noisy) to expand on implicit information in images can still enable effective reasoning and improve model capabilities.
- The paper highlights the issues arising from predicting multiple pieces of implicit information and how errors can accumulate in that setting, and also sketches a solution to the problem.

**Reasons To Reject:**

- While the paper performs multiple analyses on the proposed CLEVR-Implicit dataset, it doesn't relate the sub-problem of REC in the specific case of "same"/"different" expressions to the broader problem of REC. Do the issues with grounding these expressions persist when the model is training on a broader class of referring expressions (the answer to this is yes based on results from the CLEVR-Ref+ paper, but a more thorough study would be valuable in this context)? Does the proposed method affect performance on other types of referring expressions? If so, how?
- While the proposed method is effective on the CLEVR domain, the path to enabling this in REC with photographs is unclear. The proposed method seems to rely on components that assume access to information provided by the underlying program that generates the CLEVR image (such as an enumerable set of attributes, a templated sentence structure that can be easily manipulated to insert additional information, etc.). While expanding the scope to photographs may be beyond the scope of this work, additional discussion on how this might fit within the framework of the TIE method would improve the paper. The limitations section does discuss the role of synthetic data, but it isn't immediately clear how TIE can be made to work on more complex domains.
- Related to the above two points is the question of comparison to baselines not specifically tailed to the CLEVR-Implicit dataset. The paper is missing results that would help contextualize the difficulty of the task such as a model trained on CLEVR-Ref+ for the same task, and tested on CLEVR-Implicit. Another recent line of work [1] uses large models of code to combine the results of lower-level visual processing modules for complex visual reasoning tasks like REC. These models have been shown to be effective at a task such as REC, require no domain-specific training, and seamlessly integrate into a solution to the more generic version of the REC task. The paper doesn't clearly articulate why the approach of making the implicit information explicit might be a better approach than a modular, compositional form of explicit reasoning. This might even be a suitable baseline, but its use as one is contingent on the authors having the access and resources to closed models like Codex, or being able to get strong numbers with an open-source code model (which would be informative if possible).

[1] Sur'is, D., Menon, S., & Vondrick, C. (2023). ViperGPT: Visual Inference via Python Execution for Reasoning. ArXiv, abs/2303.08128.

**Reproducibility:**

4: Could mostly reproduce the results, but there may be some variation because of sample variance or minor variations in their interpretation of the protocol or method.

**Reviewer Confidence:**

3: Pretty sure, but there's a chance I missed something. Although I have a good feel for this area in general, I did not carefully check the paper's details, e.g., the math, experimental design, or novelty.

**Typos Grammar Style And Presentation Improvements:**

Stylistic: the space before an opening parenthesis is missing in many places (such as line 037, 047, etc.)

---

> ### Author Rebuttal · Authors · 2023-08-29
>
> Thank you for the constructive suggestions on this paper. The comments are helpful to improve our work. We have carefully studied each comment one by one. We will respond point by point to your review and questions about the paper as follow:
>
> **Response-to-Reject1**: Thank you for the constructive comment. Based on our analysis, the situation about implicit text is more likely to appear in the text which contains the same and different types. The implicit text, which hides the explicit information but can refers to an object, often emerges from the comparison between 2 or more objects. Therefore, in CLEVR-Ref+, some classes of referring expressions include: (1) directly describe the target, (2) describe the target by one or several spatial relations, (3) AND, OR and other logics etc. Existing models rely on the explicit words/information to assign the visual features so that they can do the reason regardless of the type of referring expression. But the implicit referring expressions built in CLEVR-Implicit hide the explicit information and force models to reason the implicit text directly, which cause the poor performance of current models. Besides, our proposed TIE method can generalize to other types of referring expressions, like the type of integer comparison. For example, the implicit text of integer comparison might manifest as “The middle one object in a group that have the highest count of a certain color” or more complex like “The amount of a group of objects with a certain color is two more than another group of objects with another color, which object is the left one of this group?” TIE can also build prompts for this type of examples, like “The middle one object in a group that have the largest      number of a certain color”, and then predict the amount (explicit information) of this group of objects. After transformation by TIE, current models can process this type of implicit text as well.
>
> **Response-to-Reject2**: Thank you for the constructive comment. It’s true that our generated implicit text relies on the CLEVR images and the provided annotations. And our proposed TIE method can also expand the scope beyond the CLEVR domain to the real-world photographs. For example, based on an Visual Genome [1] image (You can check the image in this anonymous link: https://anonymous.4open.science/r/example_image-C3BD/25.jpg), consider the implicit text like “The left one of two persons with the same posture and gender”. By our TIE method, prompts can be built before the “posture” and “gender”, like “The left one of two persons with the same      posture and      gender”. The prompt guides the model to predict the explicit words “walking” and “woman” like “The left one of two persons with the same walking posture and woman gender”. By integrating TIE approach, current models can also make the implicit reasoning on the real-world scenarios. Besides, thanks for your suggestion about additional discussion, we will add discussion about how the real-world scenarios fit within our TIE method.
>
> [1] Krishna R, Zhu Y, Groth O, et al. Visual genome: Connecting language and vision using crowdsourced dense image annotations[J]. International journal of computer vision, 2017, 123: 32-73.
>
> **Response-to-Reject3**: Thank you for your suggestion about the additional experiments. We have trained both MDETR and VLTVG on CLEVR-Ref+ and tested on CLEVR-Implicit, following the experimental setup in our paper. We get the results in REC task:
>
> |  | val | test |
> | :----:| :----: | :----: |
> | MDETR | 47.02 | 45.85 |
> | VLTVG | 47.99 | 47.11 |
>
> They are lower than the performance of column “Before TIE” of Table 5. This might be able to illustrate the challenge of reasoning the implicit text in our CLEVR-Implicit dataset. Moreover, we test several examples of our dataset on ViperGPT, one of the generated code is provided below:
>
> ```python
> def execute_query(image):
>     image_patch = ImagePatch(image)
>     objects = image_patch.find("object")
>     if len(objects) < 2:
>         return "Not enough objects found in the image."
>
>     reference_object = objects[0]
>     matching_objects = []
>
>     for obj in objects[1:]:
>         if obj.verify_property("object", "color") == reference_object.verify_property("object", "color") and \
>            obj.verify_property("object", "size") == reference_object.verify_property("object", "size") and \
>            obj.verify_property("object", "shape") == reference_object.verify_property("object", "shape") and \
>            obj.verify_property("object", "material") == reference_object.verify_property("object", "material"):
>             matching_objects.append(obj)
>
>     if len(matching_objects) == 0:
>         return "No matching objects found."
>
>     matching_objects.sort(key=lambda x: x.left)
>     return matching_objects[0]
> ```
> The test example is “The left object that is the same color, size, shape and material as the other one”. The generated code has two errors: (1) ViperGPT incorrectly treats the first of all objects as the reference object by default (reference_object = objects[0]), while the correct way involves traversing every two objects combination. (2) There is a problem with the search code. Although ViperGPT can understand the concept of “same” relation and attempt to find the target object by traversing the color, size, shape and material relationship between all objects and reference object, it cannot understand the implicit referring expression and regards the categories of attributes like color and size as the specific attribute values, so LLM does not understand the logic of traversal very well. Besides, existing models are pre-trained on the image-text (with explicit words) pairs so that they have limitations in processing the implicit reasoning. Our method for extracting the explicit information might be a simple but effective approach to overcome the implicit reasoning challenge of current models. And we will add the above discussion into our paper on the next modification.
>
> **Response-to-Question1**: Thank you for the constructive comment. We will add a discussion about the difference of “same” type between CLEVR-Ref+ and CLEVR-Implicit in our paper. The key difference on “same” type text between CLEVR-Implicit and CLEVR-Ref+ is whether the explicit information is hidden. The “same” type of text in CLEVR-Ref+ is like “The objects that have the same size as the red sphere”, which contains explicit information “red sphere”. This explicit context provides REC models with a reference object and helps them locate the target. But our “same” type of implicit text is like “The two objects with the same size”, which hides explicit information. Without explicit context and hard to find the reference object, REC models are forced to compare every two objects to identify what the size is and whether they are same. That is more challenging.
>
> **Response-to-Question2**: It is possible to use a simple script to add masked tokens to text, but considering the possibility of applying it to real-world scenarios in the future (with more attributes and relationships), we use a language model T5, which is also easy to train and can be extended to more complex real-world scenarios.
>
> **Response-to-Question3**: Thank you for the suggestion about this. It’s true that “After TIE” in Table 5 is the same as the “Train-pe/Val-pe” setting and “Ground-truth” is the same as the “Train-ge/Val-ge” setting. We will make this part clearer.
>
> **Response-to-Question4**: Thank you for the suggestion about prompt learning in 2.2. We will improve this section to make it closer to our proposed method.
>
> **Response-to-Question5**: Thank you for the constructive comment. We will discuss more about the “different” samples in our paper.

---

### Official Review · Reviewer_avEK · 2023-08-11

**Soundness:** 2

**Excitement:**

3: Ambivalent: It has merits (e.g., it reports state-of-the-art results, the idea is nice), but there are key weaknesses (e.g., it describes incremental work), and it can significantly benefit from another round of revision. However, I won't object to accepting it if my co-reviewers champion it.

**Paper Topic And Main Contributions:**

The paper introduces a dataset for referring expression comprehension task. The important property of this dataset is that it contains descriptions of objects (in artificial environment) which are not explicit, e.g. the object is described through some property and not mentioned explicitly. Authors describe how they built the dataset and what type of models they test and how to evaluate them on this dataset. Scores and results suggest that models rely on explicit mentions of objects as identifying objects from implicit descriptions is too hard.

**Questions For The Authors:**

1. Examples in Table 1: I would not say this is the same type of implicit texts as in the dataset that is proposed. Are these good examples then really? These require some type of external reasoning beyond image to really understand and identify the object (by understanding what makes it unique and different from others). It is really confusing to see these examples here as they are different from what authors mean by implicit descriptions (or is it?).

2. Section 3.1: so, there are 2 objects per image? Some objects are easier to identify, some harder. Can you provide more details how you generated referring expressions? Did you just combine annotations? Did you filter them in some way?

3. Line 309: MASk is added right before the attribute. This introduces structural biases. Some colors are more frequent and therefore preferred by the model and will be predicted more correctly more often. Models will learn from these biases. Do you think it's a problem?

**Reasons To Accept:**

1. It is an interesting idea: things are often not described explicitly and understanding how we can build better models and dataset for more implicit reasoning (and understanding of implicit descriptions) is important.

**Reasons To Reject:**

I think this paper has too many weaknesses to be accepted to EMNLP.

1. The main problem, I think, is that the paper does not do what it promises when it introduces the idea. Authors develop a dataset for understanding implicit descriptions, but their experiments reveal sort of expected results where models are much better in identifying objects which are described explicitly. Plus, authors actually make implicit descriptions explicit and test models on those and show that models are indeed better on explicit descriptions. This eliminates the purpose of the paper introduced in the beginning - the problem of understanding implicit descriptions. In order for it to be a better paper, experiments should focus on actually learning from implicit descriptions and not simplifying the task for the model by making descriptions explicit.
2.  A lot of relevant literature is missing. Specifically, papers on referring expression generation - NLG community has a lot of ideas which are useful for this paper. Check papers by Emil Krahmer, Ehud Reiter, Kees van Deemter, Robert Dale etc. Reading this literature and taking into account ideas about REG is needed in this paper, as, for example, it is unclear why descriptions in Figure 1 are redundant (why would you say 'the right one' as it is already half-explicit?) and REG literature has a lot of ideas about informativeness of descriptions, the order of attributes, the important of these attributes etc.
3. Line 081: there are definitely some super related datasets that have implicit descriptions/questions. For example, Drew A. Hudson, Christopher D. Manning: GQA: A New Dataset for Real-World Visual Reasoning and Compositional Question Answering. And I would also encourage authors to explicitly motivate claims like the one in Line 086-088.
4. The idea of building prompts and prompt learning makes the task very simple as these are more of a template-based methods. Simplicity in this case is not correlating with how real explicit descriptions are - they do not follow a specific template (in fact, there is a lot of variety and deviation). Literature on REG is helpful here for the authors.
5. Writing in the paper is sometimes just too complicated. Line 277-284 can be really made clearer. Description of the Base model is very confusing. Line 374: do you maybe want to say that attributes in negative examples all have different values? Line 381: only material field? Why only material field is so important?
6. Some modelling choices are only lightly mentioned, e.g. lines 319. Why T5 and what is the purpose of using T5? It was not clear to me.

I think the paper is describing results which were obvious given the setup (which is ok, but it does not make the paper exciting) and authors deviated from the initial idea in the introduction (given that I understood it correctly).

**Reproducibility:**

3: Could reproduce the results with some difficulty. The settings of parameters are underspecified or subjectively determined; the training/evaluation data are not widely available.

**Reviewer Confidence:**

4: Quite sure. I tried to check the important points carefully. It's unlikely, though conceivable, that I missed something that should affect my ratings.

**Typos Grammar Style And Presentation Improvements:**

There is a consistent formatting issue with citations, e.g. MDETR(Kamath... There must be a space between reference and what is cited.

Line 156: confusing part of the sentence

---

> ### Author Rebuttal · Authors · 2023-08-29
>
> Thanks for the constructive suggestions on this paper. The comments are helpful to improve our work. We have carefully studied each comment one by one. We will respond point by point to your review and questions about the paper as follow:
>
> **Response-to-Reject1**: Thank you for the constructive comment. We highlight our contribution is introducing a method to simulate a human-like process for extracting explicit information from implicit textual and visual context. We argee that existing models work well in explicit text, which is validated in existing approaches. Our novelty lies in the methodology we propose to acquire explicit information. The task of transforming the implicit text to explicit involves the fusion of multimodal, extending beyond the domain of text alone. We have not found the mechanism that can enable the models to directly understand the implicit text, but we simulate the process of humans reasoning about the implicit text inspired by the thinking mode of most people on implicit reasoning (extracting the explicit information from implicit text and visual context), and propose the TIE method to enable models to do the implicit reasoning. It begins by transforming implicit text into explicit text and subsequently applies REC reasoning, a process we term "implicit reasoning". We believe our method does not eliminate the purpose of our paper, which revolves around implicit reasoning. We'd also like to clarify that the mention of "understanding the implicit text" on Line 109-110 pertains to our proposed dataset's future prospects, signifying our ongoing research trajectory. We acknowledge the potential for exploring end-to-end methods for learning directly from implicit descriptions, as you've aptly highlighted. This aligns with our future research directions, and we're actively investigating such approaches. In summary, we thank you for recognizing the significance of our contributions. This review has greatly aided in refining our work, and we're committed to addressing your concerns and insights comprehensively.
>
> **Response-to-Reject2**: (1) Thank you for your suggestion, we will cite these REG related papers. It is crucial to note that while we acknowledge the relatedness of our work to REG tasks, our approach diverges significantly. Our method involves the fusion of multimodal features and the extraction of explicit information through prompt tuning and aims to tackle the challenges posed by multimodal implicit data. Our primary focus centers on the referring expression comprehension task within the context of the multimodal domain.
>
> (2) Regarding your confusion of the examples in Figure 1, we agree that the text could be considered redundant. It's important to acknowledge that redundancy in textual descriptions can indeed arise in multimodal referring expression related datasets. A strong multimodal referring expression model demonstrates the capability to reason through redundant descriptions. A similar example is from CLEVR-Ref+ [1] dataset: “The objects that are either the fourth one of the thing(s) from right or the fourth one of the thing(s) from front that are to the left of the first one of the cylinder(s) from right”, where complex spatial relationships between objects are intentionally included to elevate textual reasoning complexity, despite clearer descriptions being feasible. Another example is from RefCOCOg [2]: “Green apple on the bottom-left corner, under the lemon and on the left of the orange”. Part of description “green apple on the bottom-left corner” is sufficient for object reference, yet redundant or supplementary details are incorporated for textual challenge. In our work, the focus doesn't hinge on the specific textual form, but rather on addressing the issue of implicit text concealing explicit information. Our goal centers around unveiling this hidden information rather than the structure of the text itself.
>
> [1] Liu R, Liu C, Bai Y, et al. Clevr-ref+: Diagnosing visual reasoning with referring expressions[C]//Proceedings of the IEEE/CVF conference on computer vision and pattern recognition. 2019: 4185-4194.
>
> [2] Mao J, Huang J, Toshev A, et al. Generation and comprehension of unambiguous object descriptions[C]//Proceedings of the IEEE conference on computer vision and pattern recognition. 2016: 11-20.
>
> **Response-to-Reject3**: Thank you for bringing up the GQA dataset in the context of reasoning and visual question answering. It’s important to note that while GQA presents valuable challenges for explicit reasoning, our dataset stands apart due to its emphasis on implicit descriptions. The examples from the GQA like:
>
> “Is the bowl to the right of the green apple?”,
>
> “Is the wine glass the same color as the beer bottle?”,
>
> “Does the soap dispenser have the same material as the faucet?”.
>
> They predominantly employ explicit words like “bowl”, “green apple”, “wine glass”, “beer bottle”, “soap dispenser” and “faucet”. These explicit cues provide a foundation for models to extract information and reason within the visual context. Conversely, our implicit dataset takes a distinct approach. It comprises implicit textual descriptions generated through attribute comparisons among multiple objects in the image. Consequently, our implicit text employs terms like "object" "same color" and "shape" deliberately refraining from revealing explicit object identifiers. This design compels models to decipher the underlying explicit information concealed within the implicit text before undertaking any reasoning process. Therefore, the differences between our dataset and other datasets stem from the unique nature of our implicit descriptions. These differences underscore the complexity of our task, necessitating the development of methods capable of discerning concealed explicit information before embarking on subsequent reasoning steps.
>
> **Response-to-Reject4**: Thanks for your suggestion to explore the literature related to REG tasks. Investigating REG methodologies will contribute to further refinement of our approach. It’s important to note that the task of transforming implicit text to explicit text is not a REG task, while the input of our method contains both visual and textual modalities, and the target object is not indicated in the input, so the process is more complex than REG. We agree with you about the explicit descriptions have a lot of variety and deviation. However, we maintain that explicit reasoning lies not solely in the form, but in the presence of explicit information within the text. The ability of models to perform explicit reasoning hinges on the identification and utilization of these explicit cues. And our goal centers around unveiling this hidden information in implicit text.
>
> **Response-to-Reject5**: Thank you for pointing out the problems in the writing of the paper. We will modify the paper in the revision stage.
>
> **(1) Clarification of TIE modules and predicting strategies. (Line 277-284):**
> We introduce two modules in TIE. The prompt design module utilizes T5 model to build prompts for implicit text. While the cloze procedure module prompts the ViLT model to fill in the hidden information. We are sorry about make this confusing, and we will make this clearer. Additionally, Base approach is our predicting strategy to predict all attribute values for implicit text with multiple attributes. in contrast to Multi-step approach which involves step-wise predictions.
>
> **(2) Explanation on negative example construction (Line 374):**
> We recognize the need for a more comprehensive explanation of how negative examples are constructed. To illustrate this, consider an image containing four objects, two are red and rubber, while other two are blue, but rubber and metal. The label (explicit text) in the cloze procedure module might be “The left one of two objects with the same red color and rubber material”. When the model is predicting the value of the first attribute(color), it could yield “red” and “blue” as candidates. However, the two blue objects do not align with the second attribute (material) due to their different materials. To ensure robustness, we build the negative sample “The left one of two objects with the same blue color and [NONE] material”. This enables model to re-predict the output when encountering the [NONE] token. As aptly noted in the opinion, for the “same” type of the implicit text, part of attributes of their corresponding negative samples will have different values. It's worth noting that not all samples in our dataset have corresponding negative examples due to the constraints associated with the negative sample creation process.
>
> **(3) Explanation of [NONE] token in attribute fields (Line 381):**
> Line 381 is also the explanation of the example we mentioned in Line 376-377. As our explanation in above negative samples, we will add [NONE] token to the attributes which dissatisfy the condition of description. Material field is not important, [NONE] token may be inserted in any attributes.
>
> **Response-to-Reject6**: As our explanation in Response-to-Reject5, T5 is used to build prompts for implicit text because it is simple enough but effective.
>
> **Response-to-Reject-Summary**: We believe our work goes beyond giving the setup and describing results which are obvious. As articulated in the response to reason 1, our work is to enable models to reason the implicit text. The reasoning process is structured into two steps: the first step is to retrieval the explicit information through our TIE method with the prompt tuning and some predicting strategies. And the second step is the REC task. By splitting into two steps, our TIE method can be plug-and-play and can be integrated to other referring expression models to do the implicit reasoning, and not only the setup.
>
> **Response-to-Question1**: Thanks for the constructive comment. Our proposed dataset focuses on synthesis images, so the corresponding implicit text does not cover the objects or attributes of the real-world scenes, so we plan to incorporate the discussion of daily life cases as an additional section in our paper. Table 1 shows the cases that the implicit text is possible to appear in our daily life, and our goal is to underscore the importance of researching implicit text occurrences. And the additional section will serve to showcase the applicability of our TIE framework in real-world scenarios.
>
> **Response-to-Question2**: (1) In order to enhance the difficulty of our dataset, we have filtered and chosen the images that contains 3 or more objects. It's important to note that the variance in recognition difficulty across different implicit referring expressions is not arbitrary but is determined by the amount of explicit information that models need to extract from the implicit text.
>
> (2) To shed light on our implicit referring expression generation process, we'd like to elaborate on the method used to create the CLEVR-Implicit dataset from the existing CLEVR images and their associated annotations. These annotations contain object attributes, as well as spatial relationships between objects. For each image, we will traverse all the objects in the image by comparing whether the attribute values between two objects are the same/different each time. For example, if an image contains 4 objects, this leads to C(4,2)=6 object pair comparisons (e.g., (1,2), (1,3), (1,4), (2,3), (2,4), (3,4), number is object id). We generate the implicit text if a certain pair satisfy the condition. Suppose the (2,3) combination objects’ shape and color are all cube and blue, and guarantee that no other pairs satisfy the condition “same shape and color” for avoiding the ambiguity when doing REC, then we will generate the implicit text like “The two objects with the same shape and color”. Due to the only one target of the REC task, we must choose one of the (2,3) objects to be the target. We employ a random selection for the target based on the spatial relationship between 2 and 3. If object 2 is left of object 3 and object 2 is selected, the final implicit text is “The left one of the two objects with the same shape and color.” This consistent pattern of implicit text generation in the CLEVR-Implicit dataset ensures both the concealment of explicit information and reference of target objects in the images.
>
> **Response-to-Question3**: Thank you for addressing the concern. We will incorporate the frequency data of every attribute and their values to the Table 3. Regarding the color values, we have calculated the frequency of red, blue, green, brown, yellow, purple, gray, cyan, all of which fall within the range of 2302 to 2456. We took that into account, and balanced the number of different colors. Therefore, the frequency gap is relatively small. We believe that there is no biases that some colors are more frequent and will be predicted more correctly.
>
> Thanks for pointing out the formatting issue with citations and the confusing sentence, we will correct them in the revision stage.

---

### Meta-Review · Area_Chair_fKp1 · 2023-09-18

**Recommendation:** 4

**Metareview:**

Reviewer avEK gave some very detailed critical feedback on the framing of the paper: it only looks at a subset of types of implicit reasoning, and the connection to the implicit reasoning that humans do is a bit tenuous. The authors and this reviewer had a detailed and productive discussion, and I'm confident that the authors will incorporate some of these framing suggestions in the camera ready version of the paper.

Concerns about the generality of the dataset were also raised by Reviewer Gmcn -- "same"/"different" problems are only one type of REC, and CLEVR does not use real photographs. The author response helped address these, illustrating how to apply the TIE method on Visual Genome, and making a case that findings on the dataset, and the TIE method, are also applicable to other types of REC.

All together, I'm convinced that this work is sound. In terms of excitement and likely impact, it seems that this dataset, and the proposed method, will probably be a useful component in measuring abilities on one type of implicit reasoning.

---

### Decision · Program_Chairs · 2023-10-07

**Decision:**

Accept-Main

**Comment:**

Reviewer avEK gave some very detailed critical feedback on the framing of the paper: it only looks at a subset of types of implicit reasoning, and the connection to the implicit reasoning that humans do is a bit tenuous. The authors and this reviewer had a detailed and productive discussion, and I'm confident that the authors will incorporate some of these framing suggestions in the camera ready version of the paper.

Concerns about the generality of the dataset were also raised by Reviewer Gmcn -- "same"/"different" problems are only one type of REC, and CLEVR does not use real photographs. The author response helped address these, illustrating how to apply the TIE method on Visual Genome, and making a case that findings on the dataset, and the TIE method, are also applicable to other types of REC.

All together, I'm convinced that this work is sound. In terms of excitement and likely impact, it seems that this dataset, and the proposed method, will probably be a useful component in measuring abilities on one type of implicit reasoning.